# Effects of Agricultural Intensity on Nutrient and Sediment Contributions within the Cache River Watershed, Arkansas

Amelia K. Atwell [1,2,*] and Jennifer L. Bouldin [2]

1   Wofford College Biology Department, Wofford College, Spartanburg, SC 29303, USA
2   Environmental Sciences Graduate Program, Arkansas State University, State University, Jonesboro, AR 72467, USA
*   Correspondence: atwellak@wofford.edu

**Abstract:** Streams in agricultural lands tend to serve as a conduit for nutrient pollution. These streams are often modified and have reduced riparian zones, resulting in agriculture being the leading cause of nonpoint source pollution into streams of the United States. Eutrophication within the Gulf of Mexico has been attributed to nutrient and sediment contributions from watersheds within the greater Mississippi River Basin. One such watershed, the Cache River Watershed (CRW) located in northeast Arkansas, was assessed to determine the impacts of agricultural intensity on water quality at a local scale. The objective of this study was to determine the influence of agricultural activity on nutrient and sediment contributions to the CRW. Following American Public Health Association guidelines, physicochemical parameters, turbidity, and total nutrients (nitrogen and phosphorus) were analyzed weekly from October 2017–September 2020 at 12 subwatersheds of four varying agricultural intensities (low, low moderate, moderate high, high). Results indicate that physicochemical parameters increase (pH, conductivity, temperature) or decrease (dissolved oxygen) with increased agricultural intensity. Similarly, turbidity and total nutrients also increase (significantly for turbidity and total phosphorus) with increased intensity. Contributions of sediment and nutrients in the CRW not only influence local stream health but also contribute to hypoxia in the Gulf of Mexico.

**Keywords:** agricultural intensity; total nutrients; turbidity; nonpoint source pollution; Cache River Watershed





## 1. Introduction

In the United States, the leading cause of nonpoint source (NPS) pollution into freshwater streams is agriculture [1]. Streams draining agricultural lands tend to have increased water temperatures, erosion, sediment, and nutrient inputs when compared to similar-sized forested streams [2–4]. Since river systems act as conduits for sediment and nutrients, the increased loads in subwatersheds can be cumulative within the river system, resulting in eutrophication within the freshwater system as well as marine-receiving systems [5–9]. Eutrophication, the process by which an influx of nutrients causes algal blooms in a waterbody that subsequently die off, leading to a reduction in oxygen within the water column, has led to "dead zones" along the coastlines of the United States with the Gulf of Mexico (GOM) being the largest [10–12]. Over 70% of the sediment and nutrients entering the GOM originated from upstream agricultural sources within the greater Mississippi River Basin, including Arkansas [13].

Agriculture in Arkansas accounts for 40% of all land use, with a mixture of pastureland and row crops contributing to water quality impairments across the state and the greater Mississippi River Basin [14,15]. While pastureland is found throughout the state, row crop agriculture is primarily located along the eastern portion of the state within Mississippi River Alluvial Plain (MSRAP) Ecoregion. Excessive sediment and nutrients from the MSRAP have been attributed to NPS pollution within the state and eutrophication in the

GOM [16]. One area of particular interest for contributions of sediment and nutrients within the MSRAP is the Cache River Watershed (CRW).

The CRW, located in northeast Arkansas (Figure 1A,B), is a narrow and long watershed (29 km in maximum width and 230 km in length) that flows in a southwestern direction, originating in southeastern Missouri and terminating at the White River confluence near Clarendon, Arkansas [17]. The CRW is highly agricultural with nearly 70% of land use in row crop agriculture, encompassing eight of the top 18 rice-growing counties in the United States [18]. Due to the vast amount of agriculture in the watershed, stream segments of the Cache River, the Bayou DeView (its major tributary), and smaller tributaries have been listed as impaired for not meeting state water quality standards since the mid-2000s [19–26]. Because of the economic importance of the watershed and continued water quality impairments stemming from agriculture, it is important to investigate the impacts of agricultural intensity on sediment and nutrient contributions. Historical studies have shown elevated levels of nutrient and sediment present within the CRW; however, those studies focused on subwatershed level assessments and/or were conducted on a monthly (3 year) or weekly (10 week) basis, likely missing variations a 3year, weekly monitoring project could detect [27–30]. The objective of this study was to determine how agricultural intensity upstream of a sampling site affects physicochemical parameters and contributions of sediment and nutrients in 12 subwatersheds of the CRW with varying levels of agricultural activity.

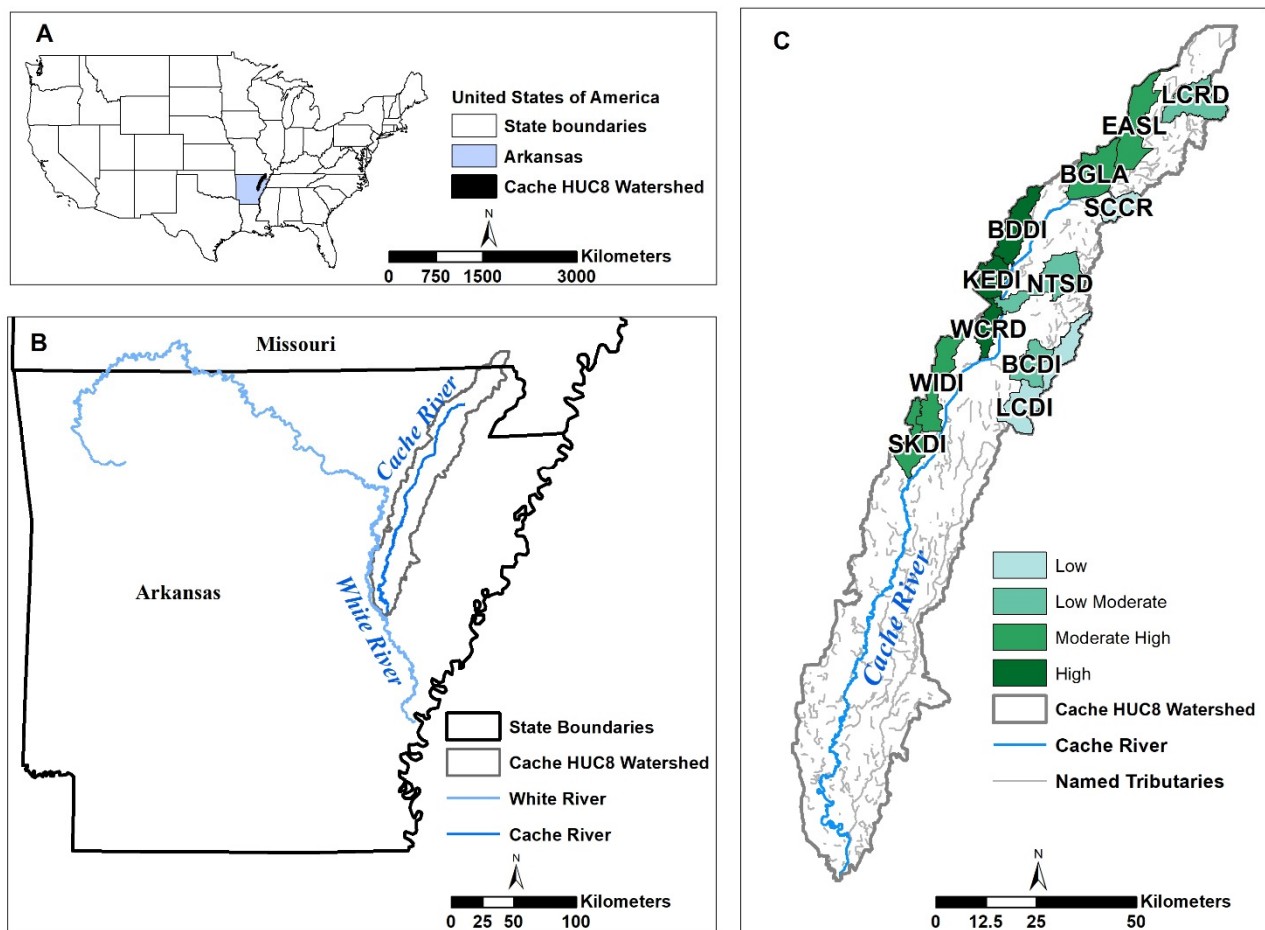

**Figure 1.** Location of the Cache River Watershed within Arkansas (**A**), the confluence of the White and Cache rivers (**B**), and agricultural intensity of sampled subwatersheds (**C**).

## 2. Materials and Methods

Physicochemical parameters (pH, conductivity, temperature, dissolved oxygen (DO)), turbidity, and total nutrients (phosphorus and nitrogen) were monitored weekly from October 2017–September 2020 at 12 subwatersheds (HUC 12) of the CRW with varying levels of agricultural intensity (Table 1). Sampling sites were located at the nearest confluence of the tributary and the Cache River and water samples were taken during a variety of water flow conditions, including both baseflow and stormflow. By using the percentage of agricultural land use above the sampling site, as determined by satellite imagery from the 2011 United States Geological Survey's National Land Cover Database, subwatersheds fell into one of four categories: low (<40%; n = 2), low moderate (41–70%; n = 3), moderate high (71–90%; n = 4), and high (>90 %; n = 3; Figure 1C) [31]. Agricultural intensity was determined using the following equation:

**Table 1.** Land use of 12 tributaries Upper Cache River Watershed, Arkansas. Site names are listed along with hydrologic unit codes (HUC), total drainage area (km$^2$), the percentage of urban, forested, agricultural drainage area upstream (US; km$^2$), non-agricultural area downstream (DS; km$^2$), calculated % agriculture US of the sampling site, and agricultural intensity classification [31]. Calculated % agriculture US was determined by the following equation: 100 × (Agricultural Land Use Area Upstream/(Total Drainage Area–Non Agricultural Area Downstream)).

| Site Name | Site Code | HUC (08020302-) | Total Drainage | % Urban | % Forested | Agricultural Area US | Non-Agr Area DS | Calculated % Agriculture US | Agricultural Intensity |
|---|---|---|---|---|---|---|---|---|---|
| Big Creek Ditch | BCDI | -0503 | 69.52 | 16.92 | 27.90 | 24.42 | 20.85 | 50.18 | Low Moderate |
| Beaver Dam Ditch | BDDI | -0207 | 100.17 | 3.10 | 0.31 | 84.91 | 8.24 | 92.35 | High |
| Big Gum Lateral | BGLA | -0202 | 117.41 | 3.37 | 0.31 | 90.52 | 14.24 | 87.73 | Moderate High |
| East Slough | EASL | -0105 | 130.89 | 4.20 | 0.04 | 65.96 | 51.90 | 83.51 | Moderate High |
| Kellow Ditch | KEDI | -0208 | 63.33 | 3.84 | 0.12 | 38.17 | 21.88 | 92.07 | High |
| Lost Creek Ditch | LCDI | -0502 | 153.31 | 17.84 | 17.13 | 27.63 | 66.40 | 31.79 | Low |
| Little Cache River Ditch | LCRD | -0102 | 105.87 | 6.48 | 24.08 | 68.99 | 1.68 | 66.21 | Low Moderate |
| Number 26 Ditch | NTSD | -0301 | 134.28 | 5.64 | 18.68 | 78.90 | 18.30 | 68.03 | Low Moderate |
| Scatter Creek | SCCR | -0601 | 50.40 | 5.91 | 66.24 | 10.77 | 1.88 | 22.12 | Low |
| Skillet Ditch | SKDI | -0401 | 76.10 | 5.85 | 0.56 | 54.46 | 12.22 | 86.80 | Moderate High |
| West Cache River Ditch | WCRD | -0303 | 48.66 | 4.05 | 0.13 | 43.89 | 0.93 | 91.95 | High |
| Willow Ditch | WIDI | -0305 | 113.71 | 2.75 | 0.55 | 64.77 | 38.86 | 86.53 | Moderate High |

Agricultural Intensity = 100 × (Agricultural Land Use Area Upstream/(Total Drainage Area–Non Agricultural Area Downstream))

Water for physicochemical measurements as well as turbidity and total nutrient analyses was collected weekly (n = 148–156) via a bucket lowered from a bridge into the thalweg, the deepest portion of the stream channel with the greatest flow. Physicochemical measurements were taken in situ using a Thermo Scientific Orion Star A329 multi-probe meter (Thermo Fisher Scientific, Waltham, MA, USA). Subsamples of approximately 45 mL (total nutrients) and one liter (turbidity) of unfiltered water were collected, stored on ice, and transport back to Arkansas State University's Ecotoxicology Research Facility for analyses. Samples for total nutrient analysis were stored in a −20 °C freezer for a maximum of 6 months and analyzed within 24 h of thawing. Analysis of total nutrients followed American Public Health Association (APHA) methods 4500-NO3F for total nitrogen (TN) and 4500-PB for total phosphorus (TP) where samples were digested prior to analysis on a Skalar San++ Flow-through Analyzer (Skalar, Buford, GA, USA) [32]. Turbidity samples were refrigerated at 4 °C and processed within 48 h using a Hach 2100P Turbidimeter (Hach Company, Loveland, CO, USA), following APHA method 2130-B.

State standard criteria for "channel altered Delta streams" physicochemical parameters and turbidity was utilized as exceedance values, while total nutrients were calculated using the 75th percentile for ecoregion waterbodies since a numeric state standard has not been established [33,34]. Means were calculated by averaging each week's sampling event for each intensity. The weekly means were then averaged for 156 weeks. Statistical analyses were performed using R and R Studio with an alpha level of 0.05 [35]. Data was tested for normality and transformations to reach normality were utilized. If normality could not be achieved, a non-parametric method was utilized. For parametric comparisons of

parameter means among agricultural intensities, ANOVA and Tukey post hoc analysis were employed, while Kruskal–Wallis with a Wilcoxson rank sum post hoc analysis was used for non-parametric comparisons. Additionally, a Pearson's correlation analysis (Spearman's rank if non-parametric) was performed to determine if any relationship existed among the parameters with agricultural intensity.

## 3. Results

Physicochemical parameters were normal; however, turbidity and total nutrients were not normally distributed and normality could not be reached via transformations. Because of this, the non-parametric Kruskal–Wallis test followed by a Pairwise Wilcoxson Sum Rank test were used to determine significant differences among agricultural intensities. All mean physicochemical parameters were within state guidelines and there was a significant increase for pH and conductivity ($\chi2 = 46.3$, df = 3, $p < 0.001$; $\chi2 = 185.4$, df = 3, $p < 0.001$, respectively) from low to high agricultural intensity, and a significant decrease in DO ($\chi2 = 80.55$, df = 3, $p < 0.001$; Table 2). Agricultural intensity was positively correlated with pH ($r = 0.608$, $p = 0.036$) and conductivity ($r = 0.910$, $p < 0.001$) and negatively with DO ($r = -0.674$, $p = 0.016$; Figure 2). Temperature did not exhibit a significant difference among agricultural intensities, nor was there a significant correlation, but there was a general increase from low to high agricultural intensity.

**Table 2.** Mean values and range for physicochemical water quality (pH, conductivity, dissolved oxygen (DO), water temperature), turbidity, and total nutrient parameters (total phosphorus (TP), total nitrogen (TN)) measured from October 2017 to September 2020 at four agricultural intensities of the Cache River, Arkansas. Means in bolded italics represent values above state criteria [33].

| Intensity | pH | Conductivity (µS/cm) | DO (mg/L) | Temp (°C) | Turbidity (NTU) | TP (mg P/L) | TN (mg N/L) |
|---|---|---|---|---|---|---|---|
| Low | 7.03 | 162 | 9.7 | 16.5 | 44.4 | ***0.285*** | 0.501 |
| (n = 2) | (6.20–8.23) | (58–460) | (4.7–13.8) | (1.0–31.0) | (4.2–460) | (0.087–1.500) | (0.151–3.600) |
| Low Moderate | 7.26 | 246 | 9.2 | 17.1 | 118.7 | ***0.563*** | ***0.988*** |
| (n = 3) | (6.23–8.32) | (61–640) | (6.0–13.7) | (1.3–31.9) | (14.7–843) | (0.162–2.711) | (0.199–6.328) |
| Moderate High | 7.24 | 291 | 8.6 | 17.4 | 203.8 | ***0.293*** | 0.565 |
| (n = 4) | (6.10–8.40) | (59–663) | (4.1–13.7) | (0.8–31.8) | (16.0–1803) | (0.124–0.922) | (0.071–3.707) |
| High | 7.32 | 329 | 8.6 | 17.4 | 170.6 | ***0.292*** | 0.631 |
| (n = 3) | (6.13–8.81) | (64–849) | (3.6–15.7) | (1.1–32.6) | (11.0–946) | (0.121–0.829) | (0.045–9.074) |

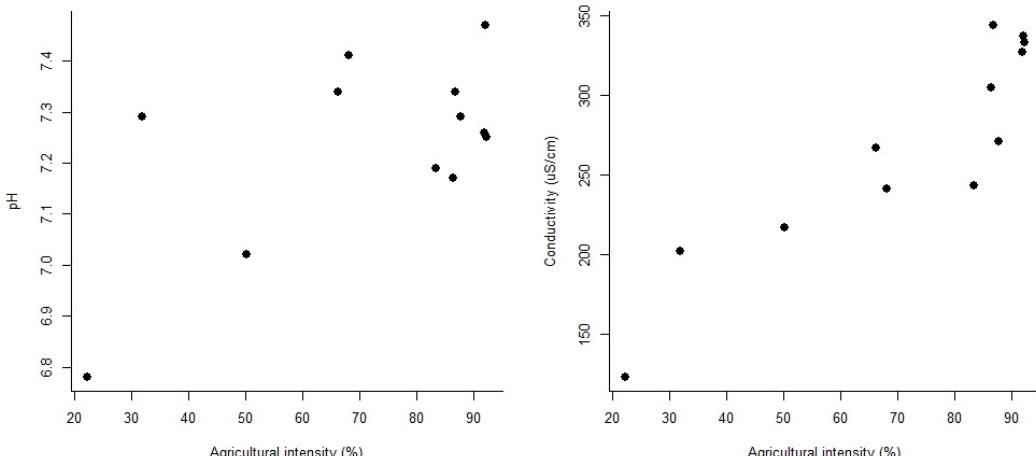

**Figure 2.** *Cont.*

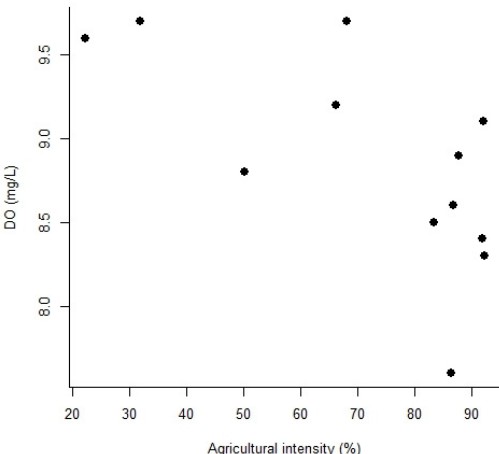

**Figure 2.** Relationship between physicochemical parameters and agricultural intensity in the Cache River Watershed, Arkansas, sampled weekly from October 2017 to September 2020.

Similarly, mean turbidity increased from low to high intensity; however, the moderate high intensity had the greatest mean nephelometric turbidity unit (NTU). Mean and median NTUs for all agricultural intensities were below the 250 NTU criteria set for "channel altered Delta" streams by the Arkansas Pollution Control and Ecology Commission (APCEC); however, significant differences existed among sites ($\chi2$ = 346.8, df = 3, $p$ < 0.001) [33]. No statistical similarities existed among agricultural intensities, as all were significantly different from one another ($p \leq 0.02$; Figure 3). Turbidity and agricultural intensity were not significantly correlated with one another (rs = 0.522, $p$ = 0.082). Throughout the 3-year sampling period, the 250 NTU criteria set by the APCEC was exceeded >20% of the time in moderate high- and high-intensity areas, below the 25% criteria.

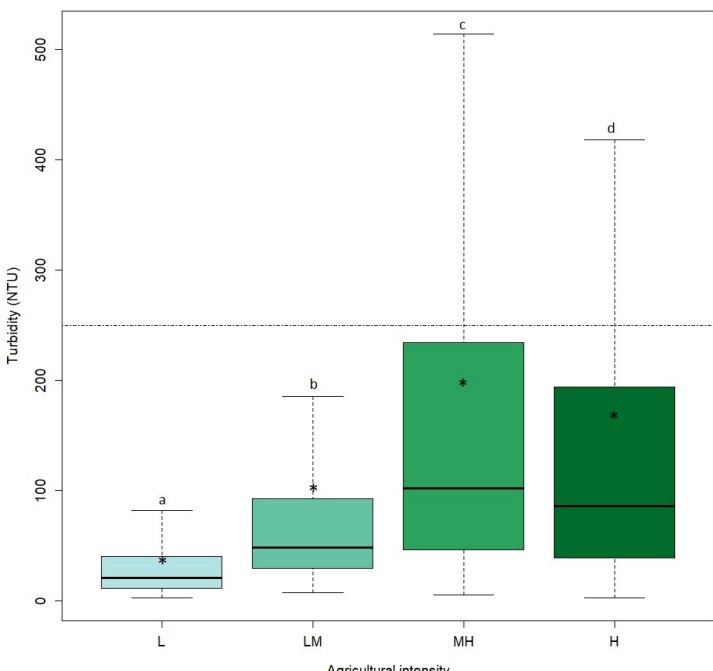

**Figure 3.** Box and whiskers plot with outliers excluded of turbidity among agricultural intensities in the Cache River Watershed, Arkansas sampled weekly from October 2017 to September 2020. The solid black line represents the median turbidity, the asterisk represents mean turbidity, the solid lines above and below the box represent the 75th and 25th percentiles, and the black dashed line represents the 250 NTU limit. Differences in lowercase letters indicate a significant difference ($\alpha$ = 0.05) among intensities.

State criteria for total nutrients, as defined by the 75th percentile for the "channel altered Delta" streams, was determined to be 0.760 mg N/L for TN and 0.240 mg P/L for TP. Significant differences existed among agricultural intensities for TN ($\chi2 = 97.14$, df = 3, $p < 0.001$) and TP ($\chi2 = 90.95$, df = 3, $p < 0.001$). The low moderate intensity had a significantly greater ($p < 0.001$) mean value of TN than any other intensity (Figure 4A). Mean TN values for the low, moderate high, and high intensities were statistically similar ($p \geq 0.85$) to one another with an increase in concentration from the low (0.501 mg N/L) to high intensity (0.631 mg N/L). Low moderate was the only agricultural intensity to have a mean TN value exceed state limits (0.760 mg N/L) and was nearly 2× greater than the low intensity. For TP, all agricultural intensities exceeded state criteria (0.240 mg P/L) and the low moderate intensity had a significantly greater ($p < 0.001$) mean value and was almost double that of any other agricultural intensity (Figure 4B). Unlike TN, the low agricultural intensity had a significantly lower ($p \leq 0.003$) mean TP value, while the moderate high and high intensities were statistically similar ($p = 0.48$) to one another. Agricultural intensity was not correlated with TN (rs = $-0.070$, $p = 0.834$) or TP (rs = 0.340, $p = 0.279$).

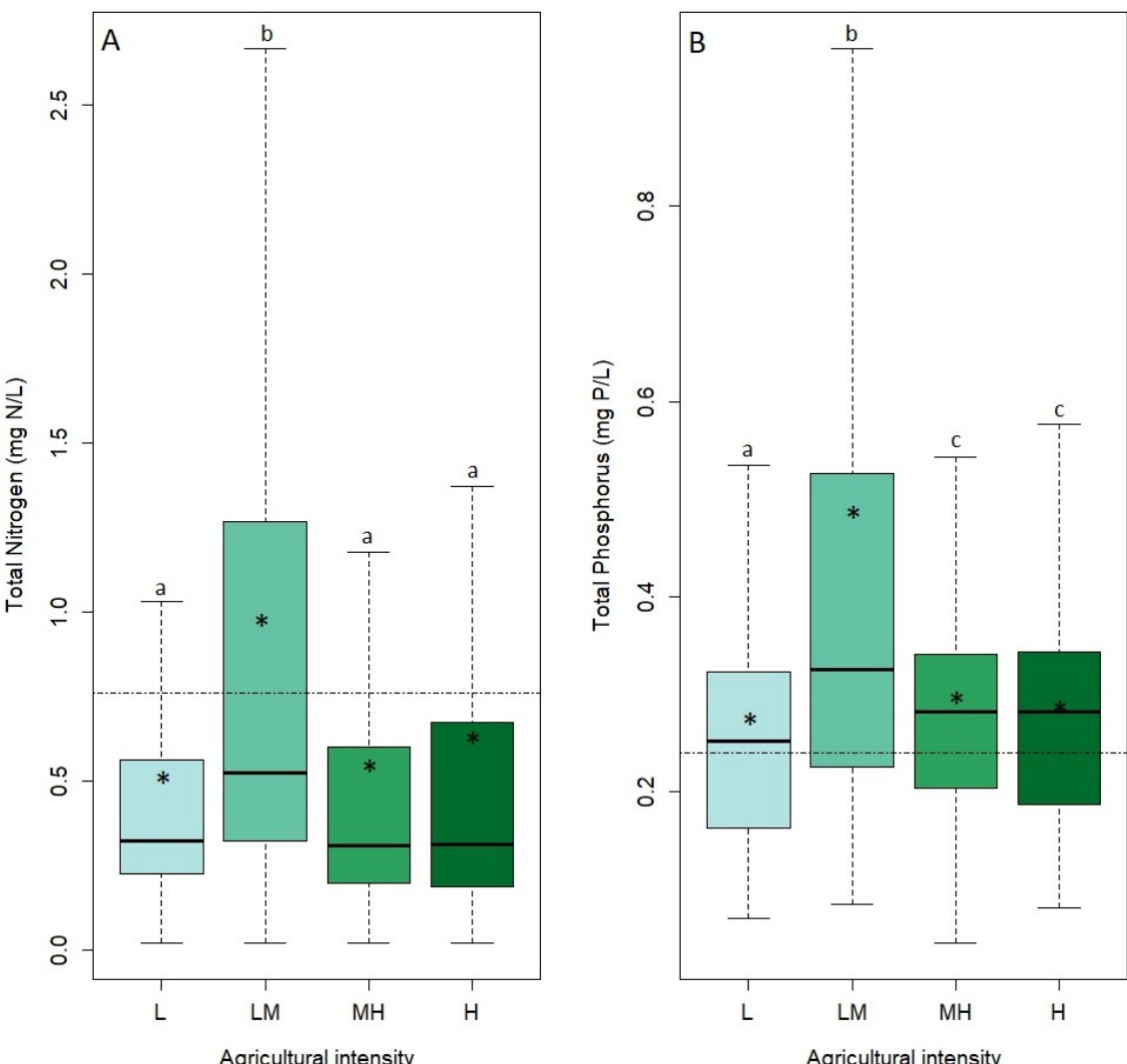

**Figure 4.** Box and whiskers plot with outliers excluded of total nitrogen (TN; (**A**)) and total phosphorus (TP; (**B**)) values among agricultural intensities in the Cache River Watershed, Arkansas sampled weekly from October 2017 to September 2020. The black dashed lines represent the state 75th percentile standard of 0.760 mg N/L TN (**A**) and 0.240 mg P/L TP (**B**) and the asterisk represents mean TN and TP. Differences in lowercase letters indicate a significant difference ($\alpha = 0.05$) among intensities.

## 4. Discussion

While within state criteria, mean physicochemical parameters exhibited a significant increasing trend, or decreasing for DO, from low to high agricultural intensity. Increases in mean pH and conductivity are likely driven by the underlying geology of the subwatersheds and subsequently the agricultural intensity, as exhibited by the positive correlations found in the present study [36,37]. Moderate high- and high-intensity subwatersheds are located entirely within the MSRAP Ecoregion while subwatersheds in the low and low moderate intensities are located partially (predominantly for low intensity) within the Crowley's Ridge Ecoregion and have less agricultural land use due to the underlying soils and topography of the land (Table 1). The MSRAP has relatively flat topography and soils are clayey and poorly drained, while Crowley's Ridge Ecoregion soils are generally well drained and loose with steeper slopes than the surrounding MSRAP [38,39]. Since the topography and soils are better suited for row crop agriculture in the MSRAP, there are likely more fertilizers (and therefore ions) entering the waterways which subsequently can lead to the increase in pH and conductivity in the moderate high and high intensities [40,41]. Though not significant, the increase in mean water temperature alongside agricultural intensity is likely due to the increase in agricultural (and subsequent decrease in forested) land use. Studies have reported that streams in forested settings have water temperatures 3–4 °C lower than those in agricultural settings [3,42,43]. This increase in water temperature is likely contributing to the significant decrease in DO from low to high agricultural intensity due to the inverse relationship between the two parameters [44].

Landscape metrics, such as agricultural intensity or land use percentage, have explained increases in not only physicochemical parameters but also turbidity, which supports results from the present study [45,46]. In the present study, mean turbidity of low-agricultural-intensity sites (with greater forested land use) was 26% less than high-intensity sites (little to no forested land), similar to results reported in Piedmont Ecoregion streams of North Carolina [47]. While the mean turbidity in the high intensity was significantly greater than the low intensity, it was also significantly lower than the moderate high intensity, which had the greatest mean turbidity. The moderate high-intensity sites were influenced greatly by one subwatershed, East Slough (EASL) with a mean turbidity of 339 NTU, exceeding state criteria and over 100 NTU greater than the next highest turbidity (Table 3). Excluding EASL establishes a mean turbidity of 158.7 NTU for the moderate high intensity. This exclusion results in a stepwise increase in mean turbidity from low to high agricultural intensity. Other factors associated with agriculture that may have led to the increase in turbidity include: dredging to clear stream bank vegetation and large woody debris for unimpeded water flow, channel modification for drainage pipes from adjacent agricultural lands, and the use of heavy equipment [48–52].

**Table 3.** Mean values and ranges for physicochemical water quality (pH, conductivity, dissolved oxygen (DO), water temperature), turbidity, and total nutrient parameters (total phosphorus (TP), total nitrogen (TN)) measured from October 2017 to September 2020 at 12 tributaries of the Cache River, Arkansas. Means in bolded italics represent values above state criteria [33].

| Site | pH | Conductivity (µS/cm) | DO (mg/L) | Temp (°C) | Turbidity (NTU) | TP (mg P/L) | TN (mg N/L) |
|---|---|---|---|---|---|---|---|
| BCDI | 7.02 (6.31–7.75) | 217 (52.7–920) | 8.8 (4.7–13.0) | 17.7 (0.6–33.2) | 56.9 (7.68–962) | ***1.032*** (0.173–4.862) | ***1.405*** (0.197–6.328) |
| BDDI | 7.25 (6.08–8.55) | 333 (46.3–799) | 8.3 (3.8–13.8) | 16.8 (0.2–30.2) | 139.5 (6.37–1070) | ***0.262*** (0.081–0.512) | 0.606 (0.088–5.729) |
| BGLA | 7.29 (5.99–8.92) | 271 (19.4–859) | 8.9 (2.1–14.0) | 16.9 (0.0–32.4) | 185.1 (9.36–1194) | ***0.285*** (0.103–0.752) | 0.569 (0.041–4.934) |
| EASL | 7.19 (5.93–8.45) | 243 (47.3–625) | 8.5 (4.2–14.7) | 16.4 (0.0–31.5) | ***339.3*** (14.1–3384) | ***0.325*** (0.107–3.158) | 0.632 (0.020–6.639) |
| KEDI | 7.47 (6.22–9.68) | 337 (71.3–1023) | 9.1 (2.8–25.7) | 17.7 (0.7–33.7) | 147.3 (2.93–1256) | ***0.307*** (0.081–1.866) | ***0.847*** (0.020–25.142) |
| LCDI | 7.29 (6.31–8.66) | 202 (40.5–725) | 9.7 (2.5–13.9) | 18.8 (1.9–36.6) | 60.4 (4.04–812) | ***0.355*** (0.101–2.736) | 0.404 (0.020–6.006) |

**Table 3.** *Cont.*

| Site | pH | Conductivity (µS/cm) | DO (mg/L) | Temp (°C) | Turbidity (NTU) | TP (mg P/L) | TN (mg N/L) |
|---|---|---|---|---|---|---|---|
| LCRD | 7.34 (6.08–9.17) | 267 (47.7–673) | 9.2 (5.7–14.7) | 15.5 (0.1–29.3) | 210.4 (17.8–2428) | ***0.290*** (0.111–0.841) | 0.739 (0.040–8.088) |
| NTSD | 7.41 (6.13–9.40) | 241 (48.6–809) | 9.7 (3.9–15.4) | 18.5 (0.7–34.1) | 93.3 (7.54–1260) | ***0.280*** (0.085–1.295) | 0.700 (0.020–5.715) |
| SCCR | 6.78 (5.93–8.46) | 123 (38.9–300) | 9.6 (5.9–14.0) | 14.2 (−0.1–26.7) | 28.4 (3.05–377) | 0.215 (0.070–0.886) | 0.598 (0.134–3.020) |
| SKDI | 7.34 (6.16–8.48) | 344 (56.3–849) | 8.6 (4.2–14.3) | 18.4 (0.9–34.0) | 151.2 (5.61–2280) | ***0.279*** (0.045–1.618) | 0.483 (0.020–3.475) |
| WCRD | 7.26 (6.10–8.66) | 327 (47.4–963) | 8.5 (1.6–15.3) | 18.1 (0.9–34.6) | 218.8 (9.80–2168) | ***0.306*** (0.124–0.678) | 0.466 (0.020–4.079) |
| WIDI | 7.17 (6.20–8.38) | 305 (65.7–762) | 7.6 (1.8–13.1) | 17.8 (0.5–31.7) | 139.8 (5.44–1824) | ***0.284*** (0.129–0.602) | 0.579 (0.060–4.868) |

For TN and TP, there was a general increase in mean concentrations from low to high agricultural intensity, though the low moderate intensity was greatest for both nutrients. With TN, the low, moderate high, and high intensities were all statistically similar with a slight numerical increase from low to high. These results support a previous study within the CRW that found unaltered sites (sites with intact riparian zones and predominantly surrounded by forested lands) had less TN than altered sites (surrounded by row crops with minimal riparian zones), though the difference was not significant [28]. Other studies have also exhibited an increase in nitrogen (TN or other forms) with increased agricultural land use [53,54]. Unlike TN, TP exhibited a significant increase from low to high intensity, again with the exception of low moderate intensity, which had the greatest mean concentration. In the greater Mississippi River Basin, over 40 % of phosphorus in the GOM have origins from row crop agriculture [13]. Decreased forested lands and increased agricultural land use, or fertilizer application on agricultural lands, also increases TP concentrations [45,55,56]. Both TN and TP had 2× greater concentrations in the low moderate agricultural intensity than any other intensity and were heavily influenced by one site, Big Creek Ditch (BCDI). BCDI has historically had elevated levels of TN and TP and is likely influenced by urban activities (i.e., the presence of a wastewater treatment plant and fertilizer application to fields at sporting complexes) within the subwatershed [57–62]. Excluding TP concentrations from BCDI results in a similar level (0.285 mg P/L) to other agricultural intensities, however, when excluding this site for TN, means remained greater (0.709 mg N/L) in the low moderate than any other intensity.

## 5. Conclusions and Future Perspectives

Agricultural intensity has a clear impact on physicochemical parameters as well as contributions of sediment and nutrients to waterways. Demonstrated by a correlative relationship, the increase in pH (7.03–7.32) and conductivity (162–329 µS/cm), with subsequent decrease in DO (9.7–8.6 °C), are most likely driven by the landscape features (soil type and topography) and alterations (riparian removal) that have enabled the increased agricultural activity at the higher intensities of this study. Although no significant correlation was found with agricultural intensity, these landscape features and alterations also influenced the significant increase in turbidity (44–171 NTU) from lower agricultural intensities. Increases in both TN and TP (0.285–0.292 mg N/L and 0.501–0.631 mg P/L) were evident as well, though only significantly for TP, indicating agricultural intensity may play a role in nutrient contributions in the CRW. Using agricultural intensity as an indicator of areas that are likely contributing more sediment and nutrients into river systems may enable a more targeted approach for future studies in the CRW as well as worldwide.

For sediment and nutrient contributions, two subwatersheds were identified as contributing more to turbidity (EASL) or TN and TP (BCDI). Further investigations into these subwatersheds, such as sampling other access points upstream of the present study sites, would be beneficial to help determine probable sources or areas of sediment and/or nutrient contributions. Subsequently, best management practices should be implemented in

problematic areas to alleviate the loss of sediment and nutrients from agricultural fields. Reductions of sediment and nutrients into the CRW would not only help to improve water quality conditions on a local scale but would ultimately reduce contributions to the GOM.

**Author Contributions:** Conceptualization, A.K.A. and J.L.B.; methodology, A.K.A. and J.L.B.; writing—original draft preparation, A.K.A.; writing—review and editing, A.K.A. and J.L.B.; supervision, J.L.B.; funding acquisition, J.L.B. All authors have read and agreed to the published version of the manuscript.

**Funding:** This research was funded by Arkansas Department of Agriculture Natural Resource Division, Workplan 17-200. Additional funding support came from Arkansas State University's Environmental Sciences Program, the Arkansas Game and Fish Commission, and Arkansas Environmental Federation.

**Institutional Review Board Statement:** Not applicable.

**Informed Consent Statement:** Not applicable.

**Data Availability Statement:** Data supporting results can be found at https://www.waterqualitydata.us/ (accessed on 1 January 2020), HUC 08020302.

**Acknowledgments:** We would like to thank the past and current students and staff at Arkansas State University's Ecotoxicology Research Facility for their assistance in the field and/or laboratory. We also thank B. Fluker, J. Harris, M. Milad, and B. Stroud for support and guidance for various aspects of the project.

**Conflicts of Interest:** The authors declare no conflict of interest.

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
