# Peer review of "Effects of Agricultural Intensity on Nutrient and Sediment Contributions within the Cache River Watershed, Arkansas"

_water, doi:10.3390/w14162528_

Round 1
Reviewer 2 Report
The manuscript has the potential to contribute to our understanding of the effects of agricultural intensity on nutrient and sediment levels. However that, many of the author's arguments require additional citations. Several examples are provided below:
1. In the introduction line number 29-30 authors mention '...Streams draining agricultural lands tend to have increased water temperatures, erosion, sediment, and nutrient inputs when compared to similar sized forested streams...' I recommend that the authors provide the following references due to justify the fact that stream networks directly related to transport sediments and nutrients: (a) Sarker et al. (2019), Critical Nodes in River Networks, Scientific Reports. https://www.nature.com/articles/s41598-019-47292-4, (b) Sarker, Shiblu, "Investigating Topologic and Geometric Properties of Synthetic and Natural River Networks under Changing Climate" (2021). Electronic Theses and Dissertations, 2020-. 965. https://stars.library.ucf.edu/etd2020/965.
2. Figures 2 and 3 can be merged to form a single figure. Please illustrate the figures using a left and right panel. Authors can gain insight from the aforementioned study. Please also superimpose the stream network of the Cache River Watershed onto the figure. On the same figure, the table 1 site names can be displayed.
3. Figure 4, 5 and 6 can be combine into one figure.
Reviewer 3 Report
Dear Authors,
The manuscript prepared by you contains valuable datasets that deserve to be published and would be of interest for decision makers, farmers, water managers, and agencies all over the world. However, it is suggested to take into consideration the specific comments provided to improve overall quality of the manuscript.
Specific comments:
1. Please check the link provided for the reference 1. United States Environmental Protection Agency (2019) as the following message appears - The procedure named attains_nation_cy.control could not be accessed, it may not be declared, or the user executing this request may not have been granted execute privilege on the procedure, or a function specified by security.requestValidationFunction configuration property has prevented access. Check the spelling of the procedure, check that the execute privilege has been granted to the caller and check the configured security.requestValidationFunction function.
2. Please check the link provided for the reference 7. Lory, J. and S. Cromley (2018) as the following message appears - Hmm. We’re having trouble finding that site.
3. Please check the link provided for the reference 8. Soil Science Society of America (2020) as the following message appears - The page you requested does not exist. For your convenience, a search was performed using the query discover soils soils in city green infra 263.
4. There is an obvious need for functionality check of all links provided in the section References.
5. In the section Introduction in the line 52 it is mentioned “…, is a 230-km long watershed…”. Kilometers as a unit of distance is not used to describe characteristics of any watershed as it does not provide any perspective on how large or small is a given watershed. Watersheds can be narrow and long, wide and long, wide and short etc. Please provide information about the area of the Cache River Watershed.
6. In Figure 2 it remains unclear what is the meaning of the thick black line. It can be assumed that it is the border of the state of Arkansas and neighboring state. In addition, the legend for this figure is missing. Please include the legend for this figure by describing all the information shown in this figure.
7. In order to understand the area of the subwatersheds monitored please add one more column in Table 1 and indicate the area of each subwatershed or describe this matter in the text.
8. If possible it would be suggested to add in Figure 3 also the streams monitored and locations of water sampling sites.
9. In the section Materials and Methods please indicate the total number of water samples analyzed as statement “…was collected weekly…” is too general.
10. Please check the requirements for the units to be used in this journal, which can be found under the tab Instructions for Authors. SI Units (International System of Units) should be used. Imperial, US customary and other units should be converted to SI units whenever possible. In the case of this manuscript please change the unites used for TP and TN from ppm to mg/L.
11. Please describe the meaning of each box, line and whisker in the text or directly in Figure 4, 5, and 6, or below these figures. Currently in the manuscript only black dashed lines and asterisks are described below Figure 4, 5, and 6 and readers have to guess what percentiles or other breaking values are used in Figure 4, 5, and 6.
Round 2
Reviewer 1 Report
No additional comments or suggestions
Reviewer 2 Report
Thanks for the revision.